# High Prevalence of Energy and Nutrients Inadequacy among Brazilian Older Adults

**DOI:** 10.3390/nu15143246

**Published:** 2023-07-21

**Authors:** Natalia Vieira Magalhães, Dan Linetzky Waitzberg, Natalia Correia Lopes, Ana Carolina Costa Vicedomini, Ana Paula Aguiar Prudêncio, Wilson Jacob-Filho, Alexandre Leopold Busse, Douglas Ferdinando, Tatiana Pereira Alves, Rosa Maria Rodrigues Pereira, Raquel Susana Torrinhas, Giliane Belarmino

**Affiliations:** 1Laboratory of Nutrition and Metabolic Surgery of the Digestive System, LIM 35, Department of Gastroenterology, Hospital das Clínicas HCFMUSP, Faculdade de Medicina, Universidade de São Paulo, São Paulo 01246-903, SP, Brazil; natalia.magalhaesnutri@hotmail.com (N.V.M.); dan.waitzberg@gmail.com (D.L.W.); apprudencio@gmail.com (A.P.A.P.);; 2Laboratório de Investigação Médica em Envelhecimento (LIM-66), Serviço de Geriatria, Hospital das Clínicas HCFMUSP, Faculdade de Medicina, Universidade de São Paulo (USP), São Paulo 01246-903, SP, Brazil; wiljac@usp.br (W.J.-F.); a.busse@fm.usp.br (A.L.B.); ferdinando@usp.br (D.F.); tatiana.p.alves@gmail.com (T.P.A.); 3Research Laboratory in Rheumatology, LIM-17, Hospital das Clínicas HCFMUSP, Faculdade de Medicina, Universidade de São Paulo, São Paulo 01246-903, SP, Brazil; rosa.pereira@fm.usp.br

**Keywords:** food consumption, energy intake, macronutrients intake, micronutrients intake, diet, food, and nutrition, older adults, sarcopenia, physical function, lean mass

## Abstract

Poor nutrition increases the risk of diseases and adverse health outcomes in older adults. We evaluated the potential inadequacy of nutrient intake among older adults in Brazil and its association with body anthropometry and composition outcomes. Dietary intake was obtained from 295 community-living older adults (>60 years old), of both genders, using a seven-day food record. Nutrient inadequacy was further identified based on the Dietary Reference Intakes and European Guidelines. Skeletal muscle mass (SM), strength and performance, and the diagnosis of sarcopenia were assessed using reference methods. Nutritional inadequacy was high, with energy, dietary fiber, and six micronutrients exhibiting the greatest inadequacy levels (>80%). Energy intake was correlated with SM strength (*p* = 0.000) and performance (*p* = 0.001). Inadequate energy, fiber, and protein intakes influenced BMI, while inadequate intake of vitamin B6 directly affected the diagnosis of sarcopenia (*p* ≤ 0.005). Further research is required to investigate whether these inadequacies can be associated with other clinical health outcomes.

## 1. Introduction

The older adult population is increasing worldwide, reaching 1 billion people in 2019 [1]. The World Health Organization (WHO) projects that this number will increase to 1.4 billion by 2030 and 2.1 billion by 2050 [1]. Unlike developed countries, low- and middle-income countries, such as Brazil, are experiencing demographic transition under unfavorable social, economic, and health conditions [2]. In 2020, 69% of older adults in Brazil lived with low income, defined as earning up to two minimum wages. This represents a serious challenge to the public health system, as the costs associated with health issues and specialized care tend to increase with age [3,4].

In this scenario, exacerbated vulnerabilities in social, physical, and mental health can occur, contributing to a loss of autonomy and increased dependency [5]. In turn, the loss of autonomy and independence amplifies functional, psychological, social, and structural alterations expected with the aging process, such as comorbidities [6], alterations in body composition [7,8], cognition [9], and food intake [10]. These changes heighten the risk of morbidity and mortality and contribute to a decline in the quality of life (QoL) of older adults [7].

Many studies have revealed poor diet quality [10,11,12] and inadequate energy and nutrient intake among older adults [13,14,15]. In Brazil, two National Dietary Surveys, conducted in 2008–2009 [16] and 2017–2018 [17], evaluated energy and nutrient intake across the population. Both surveys identified a high prevalence of inadequacy (>50%) in the intake of pyridoxine, thiamine, vitamin A, and magnesium among older adults. In the 2017–2018 survey, riboflavin inadequacy was also observed among men and older adults. The highest inadequacies (>85%) in both surveys were observed for calcium, vitamin D, and vitamin E [18].

The quality of food intake and nutrient adequacy are essential components of older adults’ QoL, as poor nutrition increases the risk of diseases and adverse health outcomes, such as sarcopenia, frailty syndrome, and disability [19]. Therefore, health professionals involved in geriatric care must be cognizant of nutritional inadequacy among older adults in their country and/or region. In this context, studies revealing nutritional inadequacies of older adults should be encouraged to support public policies and protect their health.

This study aimed to evaluate nutrient intake, its prevalence of inadequacy among older adults in Brazil, and its association with anthropometric and muscle mass variables. Our study also makes a valuable contribution to the existing national surveys by utilizing a 7-day food record (7dFR) tailored to the specific characteristics and needs of older adults [20]. The 2008–2009 survey used two 7dFR of non-consecutive days (97% response rate for the second record), while the 2017–2018 survey used 24 h recalls (24 hR) on two non-consecutive days (84% response rate for the second recall). Additionally, we conducted association analysis with muscle-related outcomes.

## 2. Materials and Methods

### 2.1. Study Design and Subjects

This was a cross-sectional study carried out in the Laboratory of Nutrition and Metabolic Surgery of the Digestive System (LIM 35), Gastroenterology Department of the Hospital das Clínicas of the University of São Paulo, School of Medicine. Data collection was carried out by the Group from the Center for the Study of Energy Expenditure and Body Composition (CEGECC) in partnership with the Laboratory of Rheumatology (LIM 17) and the Laboratory of Medical Investigation into Aging (LIM 66) of the same institution.

This study was part of a trial for the Diagnostic Evaluation of Sarcopenia in older adults. Patients registered with www.ClinicalTrials.gov (NCT04451005; 30 June 2020) and the study was approved by the local ethics committee (CaPPesq 1.905.072). All participants provided informed written consent at the beginning of the study and all protocol assessments were performed according to the Declaration of Helsinki guidelines.

Participants were recruited from the Geriatric Outpatient Clinic of the Hospital das Clínicas of the Faculty of Medicine of the University of Sao Paulo between May 2016 and December 2019. We included non-institutionalized men and women aged over 60 years who were able to answer questions asked in the anamnesis, complete screening questionnaires, and fill out the seven-day food record. We excluded elderly individuals with an amputated limb and using a prosthesis, and those with the presence of concomitant acute or chronic diseases that might interfere with the results, such as HIV, cancer, kidney, heart or lung failure, or decompensated diabetes.

### 2.2. Dietary Assessment

The dietary assessment was performed using a seven-day food record (7dFR). This is a prospective method that measures the usual intake and does not rely on memory, as foods are recorded at the time of consumption. The 7dFR data were provided by the participant or a family member. Detailed guidance was provided in order to reduce errors in reporting food and culinary preparations (Figure 1) [21,22]. To better estimate portion sizes, a manual with illustrations of different portion sizes and food models was provided [22]. Traditional kitchen utensils used to describe cooking units were also illustrated on the first page of the form provided for the 7dFR. Therefore, participants were instructed to register all beverages, food items, and culinary preparations consumed in cooking units, as described below:Culinary preparations, such as pies, cakes, and soups, were recorded in detail, with an indication of the ingredient used and the method of preparation.Processed and ultra-processed foods [23] were recorded, including the name and brand of the product.Information regarding the addition of salt, sugar, oil, and sauces was recorded, as well as information on the consumption of whole or peeled fruits and vegetables; and diet and/or light food or beverage products.Participants were asked to specify the type of juice (natural or bottled), type of milk (whole, reduced fat, low fat), type of bread and cereals (whole grain or refined grain), and type of meat (ex: chicken breast, beef chuck, pork rib).Participants received phone calls on alternating days to remind them of the 7dFR and to clarify any doubts.

#### Nutrients Intake and Prevalence of Inadequacy

The amount of energy and nutrients intake was estimated from the 7dFR using the software Virtual Nutri Plus^®^ v2.0. This software includes the Brazilian Table of Food Composition (TACO) [24] and Table of Food Composition: Support to Nutritional Decision [25]. The input was standardized, and preparations not included in the software database were added as a new recipe or item. In cases in which recipe information was lacking, we standardized the ingredients and yield from national publications [17,26], Brazilian culinary books, and websites. For processed foods, we consulted the information available on the manufacturer’s website. After standardizing recipes, we converted the cooking units to grams or milliliters [26].

The estimated nutrients were exported from the software Virtual Nutri Plus^®^ to the researchers’ database. Total energy values were expressed in kilocalories (kcal) per day, while macro and micronutrients were expressed in grams (g), milligrams (mg), or micrograms (mcg) per day. For this study, we selected the nutrients considered important to older adults [27]: carbohydrates, proteins, lipids, dietary fiber, vitamins A, B1, B2, B6, 12, D, E, niacin, and the minerals calcium, copper, iron, phosphorus, magnesium, potassium, selenium, and zinc.

The prevalence of micronutrient inadequacy was evaluated according to the Estimated Average Requirement (EAR) values proposed by the Dietary Reference Intakes (DRIs), considering the gender and age of the individuals, as shown in Table 1 [28].

For potassium intake, due to the lack of an established EAR value, we considered the Adequate Intake (AI) value [28]. Energy intake inadequacy was determined when values were lower than 30 kcal/kg for eutrophic and overweight and less than 35 kcal/kg for underweight individuals [29]. Protein inadequacy was considered when values were lower than 1 g/kg for eutrophic and overweight and less than 1.2 g/kg for underweight individuals [29]. Fiber inadequacy was considered when values were lower than 25 g/day [29].

### 2.3. Anthropometric Measurements

Anthropometric data included height and weight, measured while participants were barefoot and dressed in light clothing, using a scale accurate to the nearest 0.1 cm and 0.1 kg, respectively. Body mass index (BMI) was calculated as weight in kilograms divided by height in meters squared. All measurements were performed in duplicate, and mean values were calculated for analysis. The BMI was used to classify Nutritional Status according to Lipschitz’s (1994) [30] classification.

### 2.4. Diagnosis of Sarcopenia

The diagnosis of sarcopenia was established based on the methodology proposed by the European Working Group on Sarcopenia in Older Persons (EWGSOP2) [31]. This approach defines sarcopenia as the loss of lean mass, accompanied by diminished strength and function.

#### Muscle Mass, Strength, and Performance

The appendicular skeletal mass index (ASMI) was calculated as an indicator of skeletal muscle (SM) mass, as described elsewhere [32]. Briefly, this was determined by summing the lean mass of the four limbs, as obtained by dual-energy X-ray absorptiometry (DXA), and applying the following equation: ASMI = appendicular skeletal mass/height^2^ (kg/m^2^) [33]. SM strength was determined based on hand grip strength (HGS), which was assessed using an analog dynamometer (JAMAR^®^). During the measurement, subjects were seated in a height-adjustable chair, with their legs upright and their feet flat on the floor, to obtain a right angle in the hip, knee, and ankle joints. The test arm was kept close to the body, with the elbow flexed at a 90° angle, the palm facing the body, and the thumb pointing upwards. The arm that was not being tested remained supported and relaxed on the thigh [33,34]. SM performance was assessed by walking speed by measuring the time taken to cover a distance of 4 or 6 m [33]. Participants were instructed to fast for 4 h prior to these assessments and to abstain from physical activity (excluding light walking), diuretic use, and alcohol consumption for 24 h before these assessments.

### 2.5. Statistical Analysis

Continuous variables were expressed in summary measures (mean, median, standard deviation, and quartiles), while categorical variables were expressed in absolute and relative frequencies. The normality of continuous variables was assessed using the Shapiro–Wilk test. Correlations between energy and nutrients intake with muscle mass, strength, and performance were determined based on the Spearman coefficient. Fisher’s Exact Test was performed to assess associations between sarcopenia, nutritional classification, and polypharmacy (≥4 medications) [35,36] with energy and nutrients inadequacies. Age and gender-adjusted multivariable analyses were conducted to examine the relationship between nutrient intake and anthropometric and body composition indices, as well as sarcopenia. Multiple linear or logistic regression models were applied for continuous and binary outcomes, respectively. These models included variables that showed significant influence (*p* ≤ 0.05) on each outcome in a previous univariate analysis, also adjusted for age and gender. The stepwise forward method was used for variable selection, starting with variables with the lowest *p*-values. Each newly included variable in the multiple models had to maintain its significance without removing the significance of the variables included in the previous step. Residual analysis was conducted using the Shapiro–Wilk test, and multicollinearity was assessed using the Variance Inflation Factor (VIF). Variables with a VIF > 5 were excluded from the multiple regression model. Data were analyzed using R software (version 4.2). A significance level of 95% (*p* < 0.05) was adopted for all tests, and two-tailed hypotheses were considered.

## 3. Results

### 3.1. Participants’ Descriptive Data

The study included 295 older adults, with a mean age of 70.41 ± 7.48 years and a higher proportion of women (81.69%) compared to men (18.31%). The prevalence of overweight condition was notable within the group, while sarcopenia was present in 22.03% of the participants. Regular use of four or more medications was reported among many participants and 95% of the sample presented with at least one comorbidity (Table 1). None of the participants reported regular use of nutritional supplements.

### 3.2. Nutrients Intake and Prevalence of Inadequacy

The summary measures of nutrient intake and prevalence of inadequacy are shown in Table 2. The average daily energy intake was 1401.31 ± 452.01 kcal, representing 96.94% of inadequacy. Both protein and dietary fiber exhibited high levels of inadequacy, at 69.15% and 97.33%, respectively. Micronutrient intakes also displayed significant inadequacies. More than 50% of the participants had insufficient intake of vitamin B6, vitamin B12, phosphorus, and selenium. The greatest inadequacies among the micronutrients were observed in the consumption of Vitamin D, vitamin E, copper, magnesium, potassium, and calcium, with each showing more than 80% inadequacy among older adult participants.

When analyzing the associations between inadequacies of energy and nutrients intake according to nutritional classification, we found differences among groups (Table 3). Energy inadequacy was higher in normal and overweight participants (*p* = 0.001), while normal weight had lower inadequacies of protein intake (*p* = 0.000). Additionally, fiber inadequacy was higher among the overweight individuals (*p* = 0.005) and eutrophic participants had higher inadequacy for vitamin D (*p* = 0.028). We did not find differences in nutrients intakes inadequacies between the participants with or without sarcopenia (*p* > 0.05) (Appendix A).

A correlation between energy intake and SM strength (Rho 0.205; *p* = 0.000) and SM performance (Rho 0.189; *p* = 0.001) was observed. Likewise, fiber (Rho 0.133; *p* = 0.023) and iron (Rho 0.152; *p* = 0.009) intakes correlated with SM strength. Iron intake also correlated with SM performance (Rho 0.128; *p* = 0.028). Lastly, zinc intake correlated with ASMI (Rho 0.115; *p* = 0.048). Furthermore, Table 4 shows the multiple linear regression models adjusted by age and gender, revealing significant associations between vitamin B12, selenium with quantitative measures of skeletal muscle, specifically appendicular muscle mass, and appendicular muscle mass index (*p* < 0.01). Similarly, iron was identified as significant factor influencing handgrip strength and walking speed (*p* < 0.01), when adjusted by age and gender. Additionally, inadequate intakes of energy, total fiber, and protein, as well as inadequate intake of vitamin B6, were found to be significant factors related to BMI and the diagnosis of sarcopenia, respectively. The multiple logistic regression models confirmed the join significance of these variables for each of the respective anthropometric outcomes (Table 5). Finally, the use of polypharmacy was associated with insufficient consumption of vitamin A (*p* = 0.047) and dietary fiber (*p* = 0.016).

## 4. Discussion

Our study revealed a high inadequacy in the intake of energy, protein, fiber, and 63% of the evaluated micronutrients among older adults. These findings are noteworthy, as some of these macro- and micronutrients also exhibited correlations with relevant anthropometric and body composition outcomes. Furthermore, these inadequacies were found to significantly influence BMI and even sarcopenia, which had a prevalence of 22% within the expected range of 15.4% to 25.6% [37,38,39,40]. Overall, the participants were overweight, reported regular usage of four or more medications, and 95% had at least one comorbidity.

The nutritional inadequacies observed in this study are based on specific recommendations and guidelines tailored to older adults [28,29,41], which propose ideal values to maintain nutrients homeostasis in healthy individuals. It is important to clarify, however, that these inadequacies do not automatically equate to the manifestation of clinical health outcomes. Such outcomes are influenced by many factors, such as nutritional bioavailability and individual requirements, genetic factors, and overall nutritional status throughout life [42]. Nonetheless, considering that our participants exhibited anthropometric, health, and sociodemographic characteristics that are recognized risk factors for negative clinical outcomes [43], it is notable that nutritional inadequacies need to be considered as a contributor to these outcomes.

In this context, energy and protein inadequacies have been linked to the pathogenesis of sarcopenia and cachexia [44], and low protein intake alone with frailty status in older adults [45]. These conditions pave the way for outcomes that significantly impact the health of older adults [44]. On the other hand, our findings did not show differences in energy and protein intake inadequacies between sarcopenic and non-sarcopenic older adults. We believe that some factors contributed to these results. First, the prevalence of inadequacies was high in both groups, which did not enable us to identify differences. Second, according to EWGSOP2 sarcopenia is recognized as a muscle disease and nutritional intake is one of the wide range of factors that contribute to sarcopenia development [31].

Nevertheless, we found correlations between inadequate vitamin B6 intake and sarcopenia. Experimental evidence has shown that vitamin B6 possesses ergogenic properties, and its adequate intake positively regulates the gene expression of various factors that promote skeletal muscle growth and repair [46]. In humans, vitamin B6 intake has been inversely associated with the risk of impaired mobility [47] and directly correlated with physical performance scores and chair-rise performance [48]. Furthermore, sarcopenic older adults have been found to have lower vitamin B6 intake and higher levels of homocysteine compared to non-sarcopenic older adults [49].

Moreover, we also found correlations between energy and some nutrients with muscle mass, strength, and performance. Indeed, other studies have reported a positive correlation between energy intake and skeletal muscle mass strength [50,51,52] and its performance in older adults [51]. It has been suggested that adequate calorie intake is critical to the maintenance of muscle quality and physical independence, since insufficient calorie intake may lead to muscle catabolism [52,53]. Iron intake inadequacy was low in our study and its intake also correlated with skeletal muscle mass strength and performance. It is known that iron is essential to skeletal muscle function since it is an essential component of myoglobin and hemoglobin [54]. This indicates that aerobic and muscle oxidative capacities are greatly dependent on iron [54]. Zinc intake inadequacy was also low in our study and its intake correlated with ASMI. Other studies also found an association between zinc intake and muscle parameters [55]. Zinc is an antioxidant nutrient with the potential to prevent or delay oxidative impairment of aged muscle decline [55].

In our study, energy intake inadequacy was higher in normal and overweight participants. Although we found differences in energy intake according to nutritional classification, the prevalence of inadequacy was very high among groups. Also, energy intake does not impact body weight alone; other factors must be considered, such as gender, physical activity, presence of diseases, and metabolism [56]. Accordingly, our multivariate analyses demonstrated that inadequate energy, protein, and fiber intakes had an influence on BMI. Notably, normal weight was associated with better adequacy of protein intake. This result alerts us to greater inadequacies of protein intake in both underweight and overweight older adults.

A high prevalence of dietary fiber inadequacy was observed among our older adults. It is well documented that dietary fiber plays pivotal roles—both directly and indirectly—in various organs and systems within the human body [57]. Consequently, the physiological benefits of adequate fiber intake include lowered glycemia, blood pressure, and cholesterol levels, along with modulation of gut microbiota and intestinal transit time [58]. Thus, a pronounced inadequacy of dietary fiber could contribute to the onset of metabolic diseases and constipation [58].

In our study, we found a positive correlation between fiber intake and SM strength, while its inadequacy was associated with higher BMI and was more prevalent among overweight individuals. Accordingly, being overweight and obese has been associated with low diet quality, including low food sources of dietary fiber. Similar to our findings, a recent study showed an association between fiber intake and BMI (β: −0.08 kg/m^2^; 95% CI, −0.10 to −0.05 kg/m^2^), relative total lean mass (β: 0.69 g/kg BM; 95% CI, 0.48–0.89 g/kg BM; *p* < 0.001), and relative appendicular lean mass (β: 0.34 g/kg BM; 95% CI, 0.23–0.45 g/kg BM; *p* < 0.001) [59]. The mechanisms related to the effect of dietary fiber on SM mass have been poorly explored [59].

Another interesting finding of our investigation was a noteworthy association between polypharmacy and insufficiencies in vitamin A and fiber consumption. These findings align with a study conducted in older adults residing within a community, which also observed a link between polypharmacy and a decreased intake of fat-soluble vitamins and dietary fiber [60]. It is important to acknowledge that polypharmacy (use of multiple medications, typically ranging from two or more to 11 or more) is a widely recognized condition that can significantly affect the health of older adults [61].

Moreover, numerous studies have reported a high prevalence of vitamin and mineral inadequacies among older adults [62]. In our study, the vitamins with the highest levels of inadequacy were vitamin D and vitamin E, followed by vitamin B6 and vitamin B12. The consequences of vitamin inadequacies can be significant. For instance, vitamin D deficiency in older adults can lead to alterations in the musculoskeletal system, heightening the risks of adverse outcomes. These include increased rates of hospitalization and institutionalization, as well as a higher likelihood of falls and loss of independence [63]. However, a low intake of vitamin D does not necessarily result in a low level of this vitamin in the body, as it can also be synthesized through sun exposure [64]. This is evidenced by a health survey conducted in São Paulo city (Brazil) which revealed that nearly 100% of the population had an inadequate intake of vitamin D, even though approximately half of the individuals did not exhibit a deficiency of this vitamin [43]. In our study, vitamin D intake inadequacy was higher among the eutrophic participants. This finding is different from that found in the United States [65]. However, the prevalence of inadequacy of vitamin D was very high among all groups, limiting our interpretation of this issue.

Vitamin E deficiency, though rare, can lead to sensory neuropathy and increased erythrocyte fragility. However, there is evidence suggesting that vitamin E intake exceeding the daily recommended intake (DRI) may benefit older adults by enhancing immune function, mitigating inflammatory processes, and increasing resistance to infection [66]. Furthermore, a study conducted in China linked a low dietary intake of vitamin E with a heightened risk of incident dementia among older adults [67]. In addition, La Fata, Weber, and Mohajeri [68] concluded that vitamin E could promote healthy brain aging and potentially delay the functional decline associated with Alzheimer’s Disease. These findings underscore the potential consequences of vitamin E inadequacy and highlight the importance of adequate intake, especially among older adults.

Among the B-complex vitamins, we observed a high prevalence of inadequacy for both vitamin B6 and vitamin B12. Vitamin B6 deficiency appears to be frequent among older adults, particularly those residing in nursing homes [69]. A Norwegian study revealed that even when intake was only marginally below the recommendations, B6 deficiency still occurred [69]. The authors emphasized that decreased absorption, increased catabolism, and impaired phosphorylation of older adults enhance the requirements for this population. More recently, the NU-AGE (Elderly Population for Healthy Aging in Europe) trial demonstrated a positive correlation between Vitamin B6 intake and improved physical performance in European older adults [70].

Vitamin B12 deficiency is also frequent among older adults [71], with contributing factors that can be attributed to malabsorption, nutritional habits, and drug interference (metformin, proton pump inhibitors, drug-affected purine, and pyrimidine synthesis) [72]. A deficiency in this vitamin is predominantly a risk factor mostly for megaloblastic anemia and various neuropsychiatric symptoms. It is therefore recommended that older adults incorporate food sources rich in vitamin B12 into their regular diet to promote overall health [72].

Inadequacies in calcium and phosphorus intake were also observed among the participants. It is well recognized that older adult populations are often at high risk of calcium deficiency due to decreased dietary calcium intake and absorption, medication interactions, and changes in bone formation and strength [73]. Adequate calcium intake can reduce the risk of fractures, osteoporosis, and diabetes in certain populations [73]. It is important to note that calcium metabolism is dependent on phosphorus, vitamin D, and protein [74] all of which also showed inadequate intake among the participants in our study.

Deficiencies in selenium and copper have been associated with age-related diseases due to an imbalance between antioxidative defense and reactive oxygen species (ROS) [75,76]. Indeed, one study showed that older adult individuals with Alzheimer’s disease exhibit lower selenium erythrocyte concentrations compared to their healthy controls [77]. Consequently, an adequate intake of selenium may be beneficial for preventing age-related diseases [75]. Also, dietary copper intake has been associated with longer telomere length, an important factor in cellular aging [78].

In our study, nearly 100% of the participants demonstrated inadequate intake of both magnesium and potassium. Magnesium insufficiency is common in aging populations, even though serum levels remain constant [79]. This insufficiency primarily arises from low dietary intake, compromised intestinal absorption, and increased urinary excretion [79]. Many human diseases, such as cardiovascular diseases, Alzheimer’s disease (AD), and other dementia syndromes, as well as muscular diseases like muscle pain, chronic fatigue, and fibromyalgia, have been linked to magnesium deficits. Thus, sufficient dietary intake of magnesium is recommended to maintain a balanced nutrient profile [79]. Similarly, low potassium intake has been associated with conditions like hypertension, cardiovascular disease, chronic kidney stone formation, and low bone-mineral density [80]. Moreover, potassium is a nutrient abundant in fruits and vegetables and has been considered a marker of a healthy diet [81]. Therefore, maintaining an adequate intake of potassium is essential, particularly for the older adult population.

Our findings reveal even greater nutrient inadequacy among older adults compared to the data from the Brazilian National Dietary Surveys conducted in the years 2008–2009 [16] and 2017–2018 [17]. This discrepancy might be attributable to the different methodologies used to assess dietary habits. Additionally, a Brazilian study reported a low daily protein intake of 65.12 g/day among 47 women [82] and the Health Survey of the City of Campinas (ISACAMP) highlighted a high prevalence of inadequate dietary fiber intake (86.6%) [83].

The 2015 Health Survey of São Paulo, Brazil (ISA-Capital) [84] demonstrated that the quality of diets among adults and older adults was associated with the presence of one or more diseases such as diabetes mellitus, hypertension, cancer, or hypercholesterolemia (β = 0.636, *p* < 0.001). In this context, it is noteworthy that most of our study participants regularly used four or more medications and 95% had at least one comorbidity.

Other countries also reported a similarly high prevalence of nutrient inadequacies in older adult populations. For instance, the National Health and Nutrition Survey (Ensanut) from 2006 and 2012 indicated that Mexicans older adults exhibited dietary deficiencies in vitamins (A, B-12, C, D, and folate) and minerals (calcium, iron, and zinc) [85]. In Europe, the prevalence of inadequacies in vitamin D, folic acid, calcium, selenium, and iodine was reported to be higher than 20% among older adults [86]. In the United States, older adults with obesity showed higher micronutrient inadequacies for magnesium (both sexes), calcium, vitamin B6, and vitamin D (women only) when compared to those of normal weight [65]. However, eutrophic individuals also showed a high prevalence of inadequacies for calcium (women only) and vitamin D and E [65].

Our study does have limitations. Firstly, the relatively small sample size and the predominance of females may limit the generalizability of our findings. Secondly, it should be noted that certain comorbidities and medications can significantly influence nutrient requirements and metabolism, factors that are not considered in the available references for determining nutrient inadequacies. Lastly, we did not measure nutrients biomarkers, which means we cannot imply that nutrient intake inadequacies were associated with clinical health outcomes. Given the prevalent nutrient inadequacies found among older adults in our study, the use of specially designed oral nutritional supplements could be considered. These supplements could help counteract deficiencies and enhance overall nutritional status, potentially improving health outcomes. However, their use should be determined in consultation with a healthcare professional, considering each individual’s specific needs. Further research is necessary to evaluate the effectiveness of these interventions in the older adult population.

## 5. Conclusions

We found a high prevalence of inadequate energy, macronutrients, and micronutrients intake among older adults. Although we did not find differences in energy and nutrient intake inadequacies according to sarcopenia, the intake of energy and some nutrients was correlated to muscle mass variables. Furthermore, inadequacy of the ergogenic vitamin B6 intake was shown to influence the syndrome. Also, we found associations between inadequacies of energy and nutrients intake according to nutritional classification and polypharmacy. Further research is required to investigate whether these inadequacies can be associated with biomarkers of nutrients deficiencies and other health clinical outcomes.

## Figures and Tables

**Figure 1 nutrients-15-03246-f001:**
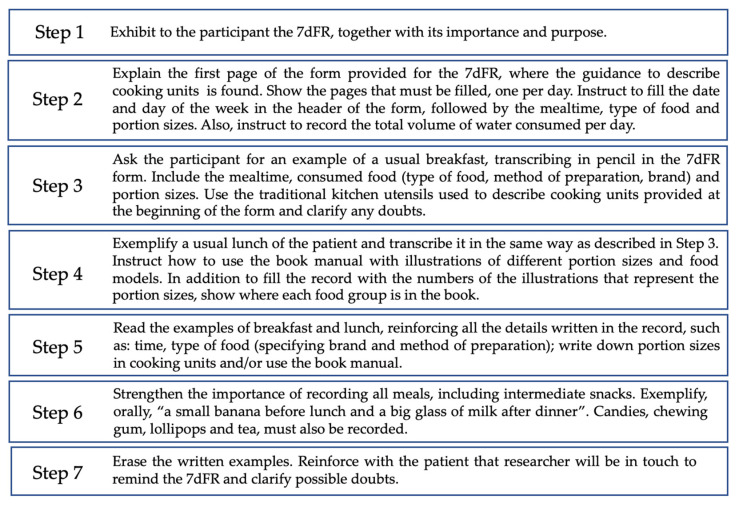
Detailed guidance to fill the seven-day food record (7dFR).

**Table 1 nutrients-15-03246-t001:** Sociodemographic and anthropometric characteristics of older adults.

Variables	Total (*n* = 295)
Age, years	70.41 ± 7.48
Gender, *n* (%)	
Female	241 (81.69)
Male	54 (18.31)
Nutritional classification ^1^ *n* (%)	
Underweight	31 (10.51%)
Eutrophic	123 (41.69%)
Overweight	141 (47.80%)
Muscle outcomes ^2^	
SMM, kg ^3^	16.43 ± 4.58
ASMI, kg/m^2 4^	6.59 ± 1.60
HGS, kgf ^5^	16.90 ± 7.00
Walking speed, m/s	1.16 ± 0.35
EWGSOP2 ^6^, *n* (%)	
Sarcopenic	65 (22.03)
Non-sarcopenic	230 (77.97)
Comorbidities, *n* (%)	
No comorbidities	16 (5.42%)
1 comorbidity	50 (16.95%)
2 comorbidities	72 (24.41%)
3 comorbidities	56 (18.98%)
4 comorbidities	47 (15.93%)
≥5 comorbidities	54 (18.31%)
Medications ^7^, *n* (%)	
no medications	29 (9.83%)
1 medication	45 (15.25%)
2 medications	38 (12.88%)
3 medications	30 (10.17%)
≥4 medications	153 (51.86%)

^1^ Body Mass Index according to Lipschitz (1994) classification. ^2^ Data are shown as mean ± standard deviation. ^3^ SMM = skeletal muscle mass. ^4^ ASMI = appendicular skeletal mass index. ^5^ HGS = hand grip strength. ^6^ EWGSOP2 = European Working Group on Sarcopenia in Older Persons. ^7^ Regular use of medications.

**Table 2 nutrients-15-03246-t002:** Nutrient intake and prevalence of inadequacy among older adults.

Nutrients	Mean ± SD ^1^	Median (Q25–Q75) ^2^	Prevalence of Inadequacy (%)
Energy (kcal)	1401.31 ± 452.01	1359.15 (1089.07–1634.00)	96.94
Carbohydrates (g)	181.30 ± 30.73	181.57 (162.09–199.21)	
Carbohydrates (%)	56.71 ± 18.80	54.47 (43.15–67.37)	
Protein (g/kg)	58.02 ± 12.60	58.81 (51.31–66.13)	69.15
Protein (%)	18.40 ± 7.19	17.46 (13.45–22.47)	
Fat (g)	41.32 ± 9.57	41.97 (36.07–46.97)	
Fat (%)	29.37 ± 11.33	28.66 (21.85–35.61)	
Dietary fiber (g)	12.54 ± 4.30	11.98 (9.59–14.78)	97.33
Vitamin A (ug)	766.35 ± 862.76	535.28 (328.4–908.94)	48.52
Vitamin B1 (mg)	1.12 ± 0.63	0.97 (0.81–1.22)	11.53
Vitamin B2 (mg)	0.86 ± 0.39	0.80 (0.60–1.05)	34.91
Vitamin B3 (mg)	13.53 ± 4.48	12.83 (10.67–16.35)	34.31
Vitamin B6 (mg)	0.97 ± 0.38	0.90 (0.70–1.21)	77.81
Vitamin B12 (ug)	3.00 ± 7.05	1.28 (0.69–2.34)	70.41
Vitamin D (ug)	1.71 ± 7.13	0.45 (0.28–0.78)	95.26
Vitamin E (mg)	8.22 ± 3.39	7.66 (6.02–9.89)	83.43
Copper (ug)	4.63 ± 34.76	0.67 (0.56–0.82)	100.00
Phosphorus (mg)	560.71 ± 149.45	542.93 (468.42–655.03)	56.50
Magnesium (mg)	134.43 ± 35.44	133.66 (110.21–156.35)	99.11
Potassium (g)	1466.10 ± 406.43	1432.96 (1182.75–1725.03)	98.22
Selenium (ug)	46.68 ± 34.09	40.81 (30.37–51.96)	58.28
Zinc (mg)	6.56 ± 3.74	5.97 (4.34–7.70)	30.47
Calcium (mg)	460.18 ± 180.04	433.02 (317.30–567.37)	98.81
Iron (mg)	11.36 ± 14.86	9.01 (7.57–10.76)	23.07

^1^ SD = Standard Deviation. ^2^ Q25 = quartile 25; Q75 = quartile 75.

**Table 3 nutrients-15-03246-t003:** Prevalence of nutrients intake inadequacies among older adults according to Body Mass Index (BMI) classification.

Nutrients	Category	Underweight	Eutrophic	Overweight	*p*-Value ^1^
Energy (kcal)	Inadequacy	27 (87.1%)	122 (99.2%)	141 (100.0%)	0.000
	Adequacy	4 (12.9%)	2 (0.8%)	0 (0.0%)	
Protein (g/kg)	Inadequacy	25 (80.6%)	66 (53.7%)	117 (83.0%)	0.000
	Adequacy	6 (19.4%)	57 (46.3%)	24 (17.0%)	
Dietary fiber (g)	Inadequacy	27 (87.1%)	119 (96.7%)	140 (99.3%)	0.005
	Adequacy	4 (12.9%)	4 (3.3%)	1 (0.7%)	
Vitamin A (ug)	Inadequacy	11 (35.5%)	57 (46.3%)	71 (50.4%)	0.318
	Adequacy	20 (64.5%)	66 (53.7%)	70 (49.6%)	
Vitamin B1 (mg)	Inadequacy	4 (12.9%)	18 (14.6%)	16 (11.3%)	0.720
	Adequacy	27 (87.1%)	105 (85.4%)	125 (88.7%)	
Vitamin B2 (mg)	Inadequacy	9 (29.0%)	40 (32.5%)	49 (34.8%)	0.833
	Adequacy	22 (71.0%)	83 (67.5%)	92 (65.2%)	
Vitamin B3 (mg)	Inadequacy	9 (29.0%)	37 (30.1%)	51 (36.2%)	0.547
	Adequacy	22 (71.0%)	86 (69.9%)	90 (63.8%)	
Vitamin B6 (mg)	Inadequacy	21 (67.7%)	89 (72.4%)	115 (81.6%)	0.091
	Adequacy	10 (32.3%)	34 (27.6%)	26 (18.4%)	
Vitamin B12 (ug)	Inadequacy	23 (74.2%)	85 (69.1%)	101 (71.6%)	0.837
	Adequacy	8 (25.8%)	38 (30.9%)	40 (28.4%)	
Vitamin D (ug)	Inadequacy	29 (93.5%)	122 (99.2%)	132 (93.6%)	0.028
	Adequacy	2 (6.5%)	1 (0.8%)	9 (6.4%)	
Vitamin E (mg)	Inadequacy	26 (83.9%)	99 (80.5%)	118 (83.7%)	0.776
	Adequacy	5 (16.1%)	24 (19.5%)	23 (16.3%)	
Copper (ug)	Inadequacy	31 (100.0%)	123 (100.0%)	141 (100.0%)	NA ^2^
	Adequacy	0 (0.0%)	0 (0.0%)	0 (0.0%)	
Phosphorus (mg)	Inadequacy	17 (54.8%)	65 (52.8%)	81 (57.4%)	0.754
	Adequacy	14 (45.2%)	58 (47.2%)	60 (42.6%)	
Magnesium (mg)	Inadequacy	30 (96.8%)	121 (98.4%)	141 (100.0%)	0.085
	Adequacy	1 (3.2%)	2 (1.6%)	0 (0.0%)	
Potassium (g)	Inadequacy	31 (100.0%)	121 (98.4%)	137 (97.2%)	0.841
	Adequacy	0 (0.0%)	2 (1.6%)	4 (2.8%)	
Selenium (ug)	Inadequacy	20 (64.5%)	61 (49.6%)	88 (62.4%)	0.078
	Adequacy	11 (35.5%)	62 (50.4%)	53 (37.6%)	
Zinc (mg)	Inadequacy	8 (25.8%)	32 (26.0%)	50 (35.5%)	0.221
	Adequacy	23 (74.2%)	91 (74.0%)	91 (64.5%)	
Calcium (mg)	Inadequacy	31 (100.0%)	121 (98.4%)	139 (98.6%)	1.000
	Adequacy	0 (0.0%)	2 (1.6%)	2 (1.4%)	
Iron (mg)	Inadequacy	9 (29.0%)	31 (25.2%)	30 (21.3%)	0.542
	Adequacy	22 (71.0%)	92 (74.8%)	111 (78.7%)	

^1^ Fisher exact test. ^2^ NA = Not applicable.

**Table 4 nutrients-15-03246-t004:** Multiple linear regression models for nutrients intake (independent variables) associated with muscle mass, strength, and performance (outcomes), adjusted by age and gender.

Nutrients	SMM ^1^	ASMI ^2^	HGS ^3^	Walking Speed
	β [95% CI]	β [95% CI]	β [95% CI]	β [95% CI]
	R^2^ (0.25)	R^2^ (0.11)	R^2^ (0.24)	R^2^ (0.10)
Vitamin B12	−0.10 [−0.16; −0.03] **	−0.03 [−0.06; −0.01] *	---	---
Selenium	−0.02 [−0.03; −0.01] **	−0.01 [−0.01; −0.00] ***	---	---
Iron	---	---	0.07 [0.02; 0.12] **	0.003 [0.001; 0.006] **

* *p* < 0.05. ** *p* < 0.01. *** *p* < 0.001. ^1^ SMM = skeletal muscle mass. ^2^ ASMI = appendicular skeletal mass index. ^3^ HGS = hand grip strength.

**Table 5 nutrients-15-03246-t005:** Multiple logistic regression models for nutrients inadequacies (independent variables) associated with body composition indices (outcomes), adjusted by age and gender.

Variable	Sarcopenia	BMI
	OR [95% CI]	OR [95% CI]
Vitamin B6 Inadequacy	2.18 [1.03–4.64] *	---
Energy inadequacy	---	14.38 [1.44–143.43] *
Fiber inadequacy	---	4.97 [1.01–24.35] *
Protein inadequacy	---	4.88 [2.06–11.58] **

* *p* < 0.05. ** *p* < 0.001.

## Data Availability

The authors confirm that the data supporting the findings of this study are available within the article.

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
