# Peer review of "High Prevalence of Energy and Nutrients Inadequacy among Brazilian Older Adults"

_nutrients, 2023, doi:10.3390/nu15143246_

Round 1
Reviewer 1 Report
In this study, the nutrient intake and prevalence of nutrient inadequacy among older adults recruited from a Geriatric Outpatient Clinic of the Hospital in São Paulo, Brazil were assessed. It is important to note that this study is descriptive in nature and does not correspond to the registration number NCT01251016 provided by the authors. The mentioned registration number refers to another trial titled "Effect of Bariatric Surgery on Gut Hormones Production and Cure of Type 2 Diabetes Mellitus." Also, the study's sample size is relatively small and the majority of the participants are women, which may limit the generalizability of the findings. Furthermore, the study lacks association analyses between nutrient intake and anthropometric indices, as well as sarcopenia. Exploring these relationships would provide valuable insights into the potential impact of nutrient inadequacy on body composition and muscle mass, particularly in older adults. Additionally, the study did not consider comorbidities (types not specified) and medications (types not specified) in the analysis, which can significantly influence nutrient requirements and metabolism. Considering the limitations and missing analyses mentioned above, it is advisable that this study is not considered appropriate for publication in the Nutrients journal.
English language needs substantial improvements.
Reviewer 2 Report
Please see attached

Just minor edits needed
Round 2
Reviewer 1 Report
I would like to thank the authors for addressing my comments.
In Table 3 the macronutrients should be presented in the order 1. Carbohydrates; 2. Protein intake; 3. Fat intake and all these expressed as percentage of daily energy intakes. In the Supplementary Table 1 only protein was added. Please also add the other macronutrients.
Regarding my comment on the association analyses between nutrient intake and anthropometric indices, as well as sarcopenia, please also perform multivariable linear regression analyses for continuous outcomes or multivariable logistic regression analyses for categorical outcomes adjusting for age and sex.
I have some minor comments:
In the Supplementary file please replace “,” with “.” For p-values.
